# Identification of Prognostic Candidate Genes in Breast Cancer by Integrated Bioinformatic Analysis

**DOI:** 10.3390/jcm8081160

**Published:** 2019-08-02

**Authors:** Charles C.N. Wang, Chia Ying Li, Jia-Hua Cai, Phillip C.-Y. Sheu, Jeffrey J.P. Tsai, Meng-Yu Wu, Chia-Jung Li, Ming-Feng Hou

**Affiliations:** 1Department of Bioinformatics and Medical Engineering, Asia University, Taichung 413, Taiwan; 2Department of Surgery, Show Chwan Memorial Hospital, Changhua 500, Taiwan; 3Graduate Institute of Biomedical Engineering, National Chung Hsing University, Taichung 402, Taiwan; 4Department of EECS and BME, University of California, Irvine, CA 92697, USA; 5Department of Emergency Medicine, Taipei Tzu Chi Hospital, Buddhist Tzu Chi Medical Foundation, New Taipei 231, Taiwan; 6Department of Emergency Medicine, School of Medicine, Tzu Chi University, Hualien 970, Taiwan; 7Department of Obstetrics and Gynecology, Kaohsiung Veterans General Hospital, Kaohsiung 813, Taiwan; 8Institute of Biomedical Sciences, National Sun Yat-Sen University, Kaohsiung 804, Taiwan; 9Division of Breast Surgery, Department of Surgery; Center for Cancer Research, Kaohsiung Medical University Chung-Ho Memorial Hospital, Kaohsiung 807, Taiwan; 10Graduate Institute of Clinical Medicine, Kaohsiung Medical University, Kaohsiung 807, Taiwan; 11National Sun Yat-Sen University-Kaohsiung Medical University Joint Research Center, Kaohsiung Medical University, Kaohsiung 807, Taiwan; 12National Chiao Tung University-Kaohsiung Medical University Joint Research Center, Kaohsiung Medical University, Kaohsiung 807, Taiwan

**Keywords:** breast cancer, weighted gene coexpression network analysis, prognosis, GEO, TCGA

## Abstract

Breast cancer is one of the most common malignancies. However, the molecular mechanisms underlying its pathogenesis remain to be elucidated. The present study aimed to identify the potential prognostic marker genes associated with the progression of breast cancer. Weighted gene coexpression network analysis was used to construct free-scale gene coexpression networks, evaluate the associations between the gene sets and clinical features, and identify candidate biomarkers. The gene expression profiles of GSE48213 were selected from the Gene Expression Omnibus database. RNA-seq data and clinical information on breast cancer from The Cancer Genome Atlas were used for validation. Four modules were identified from the gene coexpression network, one of which was found to be significantly associated with patient survival time. The expression status of 28 genes formed the black module (basal); 18 genes, dark red module (claudin-low); nine genes, brown module (luminal), and seven genes, midnight blue module (nonmalignant). These modules were clustered into two groups according to significant difference in survival time between the groups. Therefore, based on betweenness centrality, we identified *TXN* and *ANXA2* in the nonmalignant module, *TPM4* and *LOXL2* in the luminal module, *TPRN* and *ADCY6* in the claudin-low module, and *TUBA1C* and *CMIP* in the basal module as the genes with the highest betweenness, suggesting that they play a central role in information transfer in the network. In the present study, eight candidate biomarkers were identified for further basic and advanced understanding of the molecular pathogenesis of breast cancer by using co-expression network analysis.

## 1. Introduction

The incidence of breast cancer continues to increase worldwide, and currently, breast cancer is a serious disease among women. In 2017, the estimated number of new cases of invasive breast cancer was 252,710, among which 2470 men were diagnosed. In addition, 63,410 women were diagnosed with breast cancer in situ, and around 40,610 women and 460 men were expected to die of breast cancer. Presently, Asia has become a high-risk region for breast cancer, ranking first among malignant tumors in females [1,2]. Recently, with the continuous efforts and progress of modern medicine, the treatment of breast cancer has become more effective, and the mortality rate of breast cancer has been significantly reduced. However, the recurrence and metastasis of breast cancer have still not been addressed comprehensively, and have become the greatest challenges in clinical treatment [3,4].

Researchers are using genetic studies to understand the functions of tumor-related genes and roles of tumor cell signaling pathways. Therefore, we tried to understand and screen the key genes related to breast cancer recurrence to predict the recurrence factors for breast cancer accurately, in order to reduce the risk of cancer recurrence and metastasis. In addition, patients can be provided with more effective and individualized treatment, which in turn reduces breast cancer mortality and improves long-term survival.

Both bioinformatics and system biology are strong interdisciplinary fields that combine the collection, storage, processing, and dissemination of biological information, summarize life sciences and computer science, and collect and analyze genetic data [5,6].

The rapid technological breakthrough of genome-wide sequencing has provided a new perspective for the study of clinical issues and related pathological mechanisms underlying various cancers. Most studies have focused on the screening of differentially expressed genes, but not enough to understand the high degree of interconnectivity between genes, i.e., genes with similar expression patterns may be functionally related [7]. In this study, we used coexpression analysis as a powerful technique for constructing free-scale gene coexpression networks. Weighted gene coexpression network analysis has been widely used to analyze large-scale data sets and find modules of highly correlated genes. Weighted gene correlation network analysis (WGCNA) [8] has been successfully used to evaluate the associations between gene sets and clinical features and identify candidate biomarkers. Therefore, we described the correlation patterns among genes using a systems biology method based on WGCNA and identified novel biomarkers associated with breast cancer prognosis. We used WGCNA to examine a previously published dataset (GSE48213, [9]). This dataset is a transcriptional profile of a breast cancer cell line based on 56 samples used to identify the gene expression patterns associated with subtype and response to therapeutic compounds.

Finally, we used three databases, Kyoto Encyclopedia of Genes and Genomes (KEGG), Gene Ontology, and The Cancer Genome Atlas (TCGA), and completely constructed related genes and coexpression networks. We also identified key genes to facilitate further studies into their potential as biomarkers as well as therapeutic targets in breast cancer. We performed a comprehensive bioinformatics analysis to further investigate the functions, pathways, regulatory mechanisms, and drug candidates of these genes.

## 2. Materials and Methods

### 2.1. Dataset Download

The pipeline of this study is presented in Figure 1. The gene expression profiles of GSE48213 were stored in the Gene Expression Omnibus (GEO) database. It is a transcriptional profiling of a breast cancer cell line based on 56 samples associated with subtypes (basal, claudin-low, luminal, nonmalignant, and unknown).

### 2.2. Data Preprocessing

Firstly, normalized data were downloaded from the GEO database, and then we converted each Ensembl gene to Ensembl Gene ID with the biomaRt package [10,11] in R. However, a large number of these genes did not have different expression between samples. Therefore, the data was processed by focusing on the 4655 most variant genes for WGCNA analysis by using a robust method called median absolute deviation (MAD) [12]. The remaining genes were not used for analysis, because those showed no or very low changes in expression between different samples.

### 2.3. Gene Coexpression Network Construction

The expression data profiles of these 4655 genes were constructed to a scale-free coexpression network using the WGCNA package in R [8,13].

The adjacency coefficient *a_ij_* was calculated as follows: $$s_{ij}=\left|cor\left(x_{i},x_{j}\right)\right|a_{ij}=S_{ij}^{\beta }$$
where $x_{i}$ and $x_{j}$ are vectors of expression value for gene i and j. $s_{ij}$ represents the Pearson’s correlation coefficient of the two vectors. $a_{ij}\;$is the adjacency coefficient between gene i and gene j. In the presented study, the power of β=6 (scale free $R^{2}=0.88$) was selected as the soft-thresholding parameter to ensure a scale-free network. In the coexpression network, genes with high absolute correlations were clustered into the same module. The WGCNA method not only considers the association between two connected genes, but also takes associated genes into account. Modules were also identified via hierarchical clustering of the weighting coefficient matrix. To further identify functional modules in the coexpression network with these 4655 genes, the topological overlap measure, representing the overlap in shared neighbors, was calculated using the adjacency coefficient. It calculates the weighting coefficient $W_{ij}$ from $a_{ij}$ as follows: $$W_{ij}=\frac{\sum_{k=1}^{N}a_{ik}\cdot a_{ij}+a_{ij}}{\min\left(k_{i},k_{j}\right)+1-a_{ij}}$$
where a is the weighted adjacency coefficient given $a_{ij}$ and β=6 is the soft thresholding power. According to the $W_{ij}$ measure with a minimum size (gene group) of 30 for the gene dendrogram, average linkage hierarchical clustering was conducted, and genes with similar expression profiles were classified into the same gene modules using the DynamicTreeCut algorithm.

In this study, an appropriate soft threshold power (soft power = 6) was selected to construct a standard scale-free network using the pickSoftThreshold function [14,15]. Then, the network construction and module detection were performed by the one-step function “blockwiseModules” [15]. It can handle the splitting into blocks automatically, and just needed to specify the largest number of genes that can fit in a block. The R function of blockwiseModules has many parameters; parameters were implemented with the following: power = 6, maxBlockSize = 6000, minModuleSize = 30, and networkType = “unsigned” in our study.

### 2.4. Identification of Clinically Significant Modules

To identify key modules that are significantly associated with clinical subtypes of breast cancer, the expression profiles of each module were summarized by the module eigengene (ME) and the correlation between the module and clinical subtypes was calculated. The associations of individual genes with each subtype (basal, claudin-low, luminal, and nonmalignant) were quantified by Gene Significance (GS) value. Module significance (MS) refers to the average GS of all genes in the module. Modules with an absolute MS ranking first or second in all modules are considered candidates relevant to clinical traits. Finally, the module highly correlated with clinical subtypes of breast cancer was selected for further analysis.

### 2.5. Candidate Hub Genes Identification

Hub genes in the coexpression network were highly interconnected with nodes in a module, and they have been shown to have significant function. In this study, the connectivity of genes was measured by the absolute value of the Pearson’s correlation (module membership > 0.8). The module membership (MM) was defined as the correlation of gene expression profile with module eigengenes (MEs). Also, we identified hub genes in modules that were highly related to certain clinical traits, which were measured by the absolute value of the Pearson’s correlation. Thus, the intramodular hub genes were chosen based on GS (gene significance) > 0.2 and MM > 0.8. The common hub genes in the coexpression networks were regarded as “real” hub genes for further analysis.

### 2.6. Network Analysis and Visualization

Gene coexpression modules identified by clustering are often large; therefore, it is important to identify which gene in each module best explains its behavior. A widely used approach is to identify highly connected genes in a coexpression network, called hub genes [16]. In this study, we identified hub genes by betweenness centrality. Genes with high betweenness centrality are important as shortest-path connectors through a network [17]. Connectivity is often used to measure network robustness and indicates how many genes need to be removed from the network before the remaining genes are disconnected.

The hub genes detected by WGCNA can be analyzed with other tools. We performed network analysis by using the R package tidygraph (Version: 1.1.2; CRAN: MIT, MA, USA). The node centrality in networks is useful to detect genes with important functional roles. Specifically, the weighted degree of a node was used to identify potential biologically meaningful genes based on betweenness centrality [18].

### 2.7. Pathway Enrichment Analysis and Gene Ontology

To investigate a comprehensive set of functionally annotated hub genes, Gene Ontology term enrichment analysis and Kyoto Encyclopedia of Genes and Genomes (KEGG) pathway analysis were performed by using “FunRich”. FunRich is a functional enrichment and interaction network analysis tool. GO enrichment analysis and KEGG pathway analysis were performed with the FunRich functional enrichment analysis tool (version 3.1.3; ExoCarta: Victoria, Australia). The top ten terms with *p* < 0.05 and more than two enrichments were selected [19].

## 3. Results

### 3.1. Weighted Coexpression Network Construction and Key Modules Identification

Fifty-six samples with subtype data were included in the coexpression analysis. The samples of GSE48213 were clustered using the average linkage hierarchical clustering method (Figure 1). In this study, the power of β = 6 (scale-free *R*^2^ = 0.88) was selected as the soft thresholding parameter to ensure a scale-free network (Figure 2).

A total of 27 modules were identified based on average linkage hierarchical clustering (Figure 3a). Furthermore, module eigengenes (MEs) of the black, dark red, brown, and midnight blue modules were found to have the highest correlation with the subtypes (basal, claudin-low, luminal, and nonmalignant, respectively; Figure 3b and Figure 4). These modules were selected as clinically significant modules for the identification of hub genes.

### 3.2. Identification of Candidate Genes with High Weighted Degree Score

Highly connected hub genes were defined by module connectivity (module membership > 0.8) and clinical trait relationship (gene significance > 0.2). Next, the hub genes of each module were visualized as networks by using the R package tidygraph, and the top candidate genes were screened and ordered in ranks by using weighted degree scores for further analysis (Figure 5).

### 3.3. Pathway Enrichment Analysis and Gene Ontology

The genes in the clinically significant modules were categorized using the FunRich tool. Findings with higher scores are more significant than those with low scores. The GO database provides information on the associated biological processes, molecular functions, and cellular components. The GO analysis results revealed that the black module (basal) genes in the BP group were mainly enriched in cell communication, signal transduction, and cell growth and/or maintenance; the genes in the MF group were mainly enriched in molecular function unknown, translation regulator activity, and receptor activity; the genes in the CC group were significantly enriched in cytoplasm, exosomes, and nucleus. The brown module (luminal) was enriched in the BP, including regulation of nucleobase, nucleoside, nucleotide, and nucleic acid metabolism. The genes in the MF group were mainly enriched in molecular function unknown and transcription factor activity. The genes in the CC group were mainly enriched in cytoplasm, nucleus, and plasma membrane; the dark red module (claudin-low) genes in the BP group were mainly enriched in cell growth and maintenance. The genes in the MF group were mainly enriched in structural constituent of cytoskeleton and extracellular matrix structural constituent. The genes in the CC group were enriched in extracellular, cytoplasm, and extracellular space; the midnight blue module (nonmalignant) genes in the BP were mainly enriched in protein metabolism. The gene in the MF was enriched in calcium ion binding. The genes in the CC group were enriched in centrosome, exosomes, cytosol, and nucleolus. The hub genes were also over-represented in these top KEGG pathways: the black module (basal) included the integrin family cell surface interactions and beta-1 integrin cell surface interactions; brown module (luminal), mesenchymal-to-epithelial transition; dark red module (claudin-low), signal transduction; and midnight blue module (nonmalignant), peptide chain elongation. Among the functional and pathway enrichment analyses, cell division and cell cycle were the most significantly enriched.

### 3.4. Identification and Validation of Hub Genes

Based on the cut-off criteria (|MM| > 0.8 and |GS| > 0.2), 42 genes with high connectivity in the four modules (including basal, luminal, claudin-low, and nonmalignant) were identified as hub genes, and centrality measures, mainly betweenness centrality, were used. Hub genes with high betweenness centrality are important as the shortest path connectors through a network. Connectivity is typically used to measure network robustness and indicates how many genes need to be removed from the network before the remaining genes are disconnected (Figure 6). Moreover, based on the TCGA data, the expression levels of these eight genes were significantly higher in breast cancer tissues. As tumor progression always affects tumor prognosis, we also validated the eight hub genes by investigating their roles in breast cancer prognosis, including overall survival time (Figure 7).

## 4. Discussion

The progression and prognosis of breast cancer are quite variable. Although many prognostic models have been proposed, most are based on clinical parameters and lack accuracy. In the era of precision medicine, there is an urgent need for better cancer-specific prognosis and progression biomarkers to provide accurate clinical information that could significantly enhance decision-making for patient management [20].

This study is based on the gene expression profile of breast cancer, and our main aim was to analyze the molecular mechanisms underlying breast cancer. We analyzed the data from both single- and multigene layers to find differential genes involved in breast cancer development.

(1)Construction and analysis of the gene coexpression network

First, we used WGCNA, which is commonly used to construct gene coexpression modules, to explain the mechanism of breast cancer on a multigene level [21,22]. The coexpression module aggregates genes with similar expression levels into one class, but the biological pathways involved in these modules and the changes in expression levels are unknown. To address these problems, we mainly performed differential correlation analysis, functional enrichment analysis, and differential gene correlation analysis of the modules [23]. First, the module was subjected to differential correlation analysis. The gene expression spectrum was used to analyze the differences in the average expression levels of the genes in the module. Simultaneously, displacement analysis was used to calculate the false discovery rate (FDR) and four modules with significant differences in expression were found [24,25]. Next, functional analysis of the significantly different modules was performed, and the biological significance of each module was analyzed to evaluate the biological processes and functional pathways involved in breast cancer. This study is based on the gene expression profile of breast cancer to analyze the molecular mechanism underlying breast cancer. We analyzed the data from both single- and multigene layers to find differential genes involved in breast cancer development.

(2)Screening of differential genes

In this study, we downloaded the counts for breast cancer RNA sequencing and fragments per kilobase per million mapped reads gene expression data from the TCGA cancer database [22,26]. By using the cluster analysis method to remove the outlier samples, genes were restricted to protein-encoding genes and the data were regularized for preprocessing and constructing an accurate gene expression profile. Next, the differential gene analysis method was used to screen differential genes, which directly revealed the relationship between gene expression changes and breast cancer development. Our results showed a high coincidence rate with the differentially expressed genes associated with breast cancer. Noncoincident parts included the genes *TXN*, *ANXA2*, *TPM4*, *LOXL2*, *TPRN*, *ADCY6*, *TUBA1C*, and *CMIP*.

WGCNA was performed to identify the gene coexpression modules associated with breast cancer progression. The modules were identified, and several central genes were obtained from the modules. Therefore, *TXN*, *ANXA2*, *TPM4*, *LOXL2*, *TPRN*, *ADCY6*, *TUBA1C*, and *CMIP* in the common networks may be the key genes, indicating that they are highly correlated with clinical features and important biological processes.

The eight genes predicted by us have been reported for other cancers, but not for breast cancer. *ANXA2* is a member of the annexin family. Members of this calcium-dependent phospholipid-binding protein family play a role in the regulation of cell growth and signal transduction pathways. *ANXA2* has been reported to be associated with cervical, gastric, colorectal, ovarian, and liver cancers [27,28].

*TPM4*, a gene that encodes a member of the tropomyosin family of actin-binding proteins, is involved in the contractile system of streak and smooth muscles as well as the cytoskeleton of nonmuscle cells. It has been reported to be associated with lung cancer metastasis and cervical and ovarian cancers [29,30,31].

*LOXL2* is a gene that encodes a member of the lysyl-oxidase gene family. Prototype members of this family are essential for the biosynthesis of connective tissue, which encodes an extracellular copper-dependent amine oxidase. *LOXL2* can also confer additional effects on developmental regulation, aging, tumor (prostate and liver cancer) inhibition, cell growth control, and chemotaxis to each member of the family [32,33].

The locus *TPRN* encodes a sensory epithelial protein. Mutations of this locus are associated with autosomal recessive deafness. To date, few cancer-related functions of *TPRN* have been reported. It has great potential as a biomarker for breast cancer.

*ADCY6* encodes a member of the adenylate cyclase protein family, and it is required for the synthesis of cAMP. *ADCY6* promotes increased phosphorylation of various proteins, including *AKT*. It plays a role in regulating Ca^2+^ uptake and storage in the cardiac sarcoplasmic reticulum and is required for normal ventricular contraction [34].

TUBA1C, a tubulin, is the main component of microtubules. It binds two molecules of GTP, one at the exchangeable site on the beta strand and the other at a nonexchangeable site on the alpha chain. The α-2-HS-glycoprotein precursor and tubulin β chain may be involved in the pathogenesis of colorectal cancer and serve as potential biomarkers for serological diagnosis [35].

*CMIP* is a DNA-binding, leucine-containing transcription factor that acts as a homodimer or heterodimer. It plays a role in the T-cell signaling pathway. Depending on the binding site and binding partner, *CMIP* may encode a transcriptional activator or repressor protein. This protein plays a role in the regulation of several cellular processes, including cell development in embryonic lens, increased T-cell susceptibility to apoptosis, and chondrocyte terminal differentiation. *CMIP* has been reported to be associated with gastric cancer and glioma [36,37].

To further study the mechanisms underlying regulation of tumorigenesis, we performed GO and KEGG pathway and GSEA analyses. We found that the hub genes showed significant differences in cell membrane, cytoskeleton, immune cells, and ion regulation. KEGG analysis revealed that cell structure is the most important pathway (Table 1). Interestingly, we found that the central genes are often rich in cellular structures and ion signaling pathways (Appendix A). Previous studies have demonstrated that calcium regulates the cytoskeleton, and thus promotes the migration and metastasis of cancer cells [38]. Therefore, the eight hub genes may be involved in the progression of breast cancer and may influence its prognosis through calcium-regulated immune and molecular signaling pathways, contributing to the poor prognosis of breast cancer.

## 5. Conclusions

Our study used weighted gene coexpression analysis to construct a gene coexpression network and to identify and validate network hub genes associated with the progression and poor prognosis of breast cancer. Eight hub genes, namely, *TXN*, *ANXA2*, *TPM4*, *LOXL2*, *TPRN*, *ADCY6*, *TUBA1C,* and *CMIP*, were identified and validated as associated with the progression and poor prognosis of breast cancer.

## Figures and Tables

**Figure 1 jcm-08-01160-f001:**
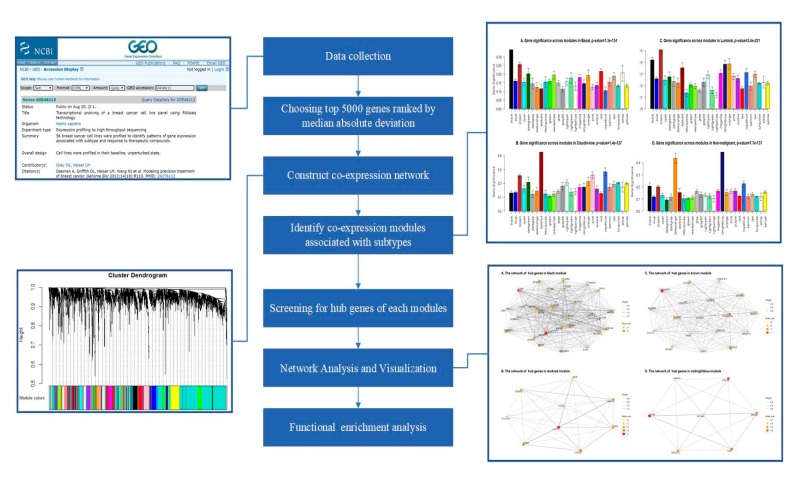
Flowchart of the study’s design, illustrating data preparation, preprocessing, and analysis.

**Figure 2 jcm-08-01160-f002:**
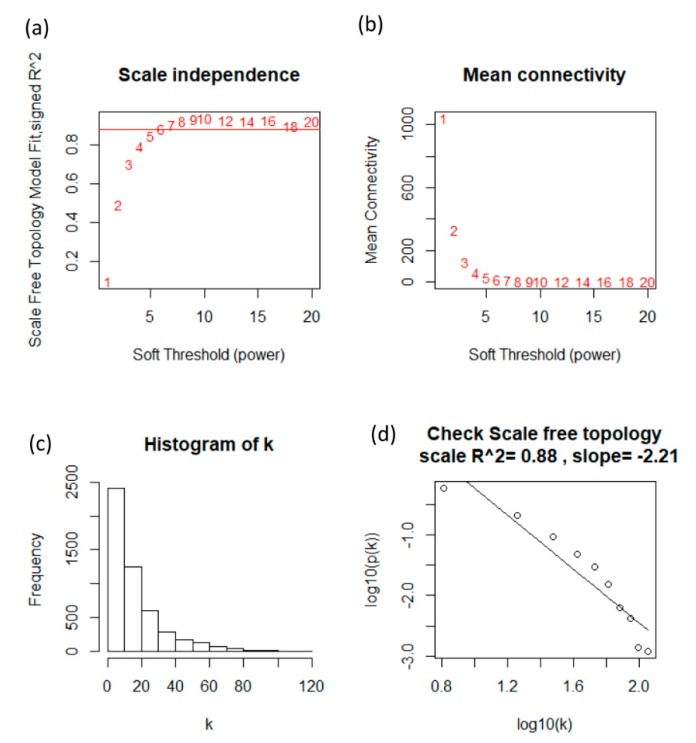
Determination of soft-thresholding power in WGCNA analysis. (**a**) Analysis of the scale-free fit index for various soft-thresholding powers β. (**b**) Analysis of the mean connectivity for various soft-thresholding powers. (**c**) Histogram of connectivity distribution when β = 6. (**d**) Checking the scale free topology when β = 6.

**Figure 3 jcm-08-01160-f003:**
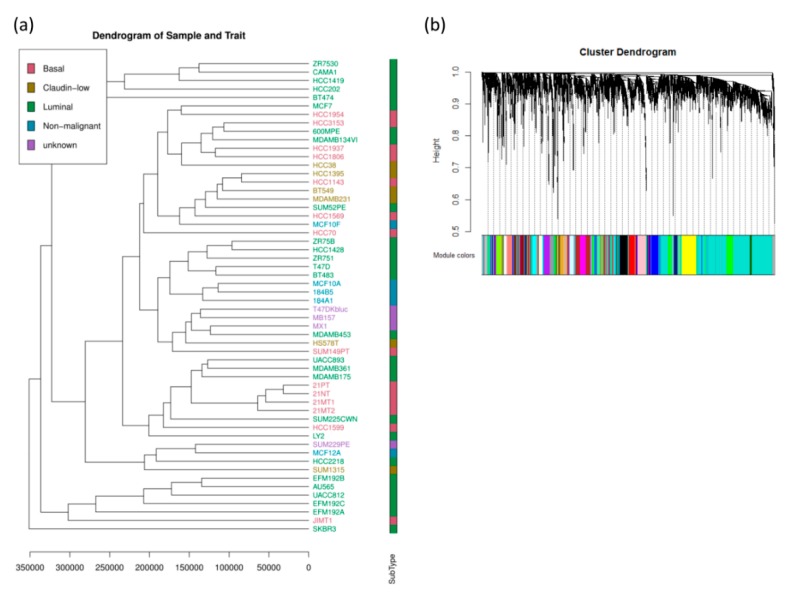
(**a**) Clustering dendrogram of 56 breast cancer cell lines and the clinical trial. (**b**) Dendrogram of all differentially expressed genes clustered based on a dissimilarity measure. Dynamic tree cutting was applied to identify modules by dividing the dendrogram at significant branch points. Modules are displayed with different colors in the horizontal bar immediately below the dendrogram.

**Figure 4 jcm-08-01160-f004:**
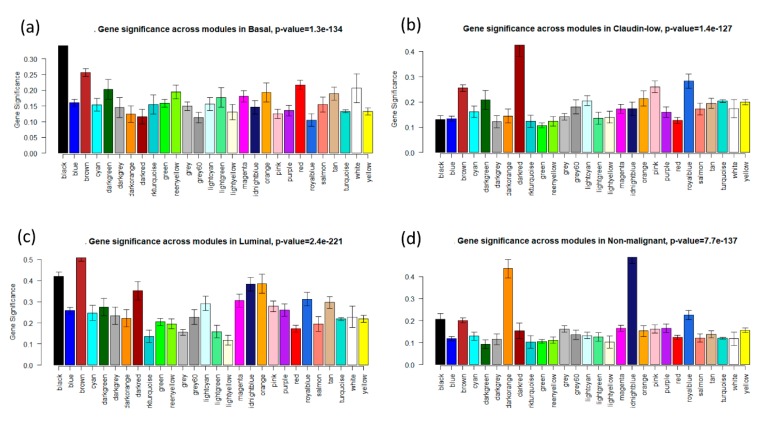
Distribution of average gene significance in the modules based on different subtypes. (**a**) Black modules—highest association with basal. (**b**) Dark red modules—association with claudin-low. (**c**) Brown modules—highest association with luminal. (**d**) Midnight blue modules—highest association with nonmalignant.

**Figure 5 jcm-08-01160-f005:**
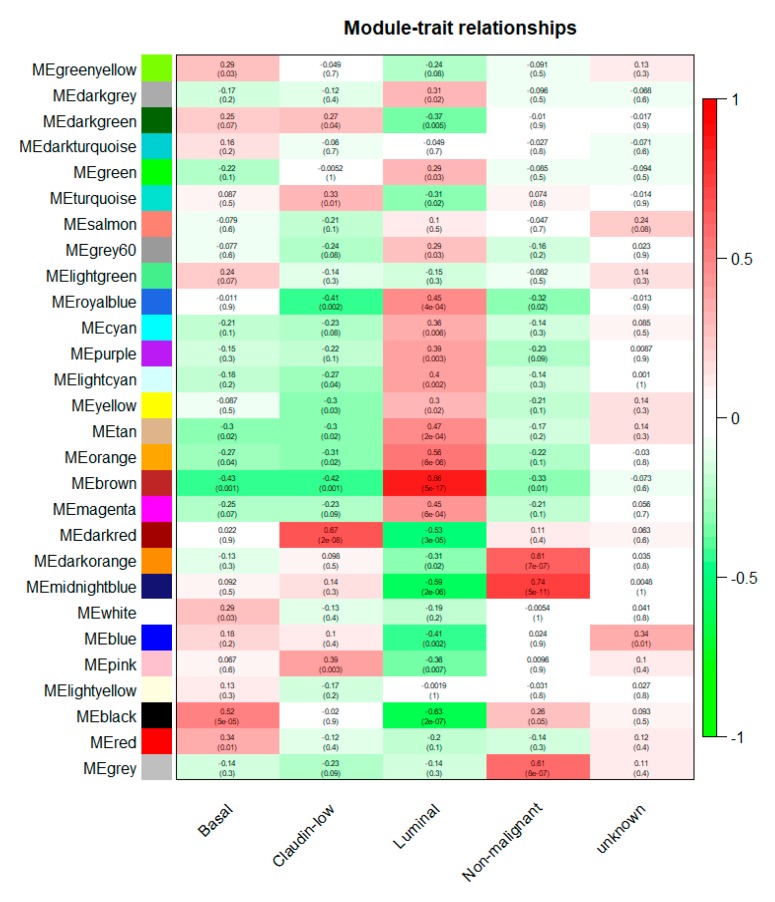
Heatmap of the correlation between module eigengenes and clinical subtypes of breast cancer. Each column contained the corresponding correlation and *p* value.

**Figure 6 jcm-08-01160-f006:**
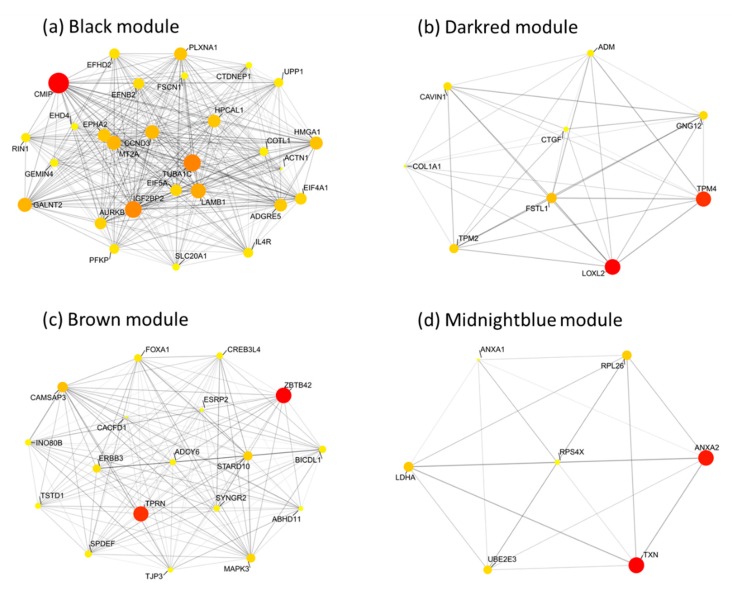
The network of hub genes in the (**a**) black module, (**b**) darkred module, (**c**) brown module, and (**d**) midnight blue module. Nodes represent genes and node size indicates weighted degree score. Edges are colored by weight.

**Figure 7 jcm-08-01160-f007:**
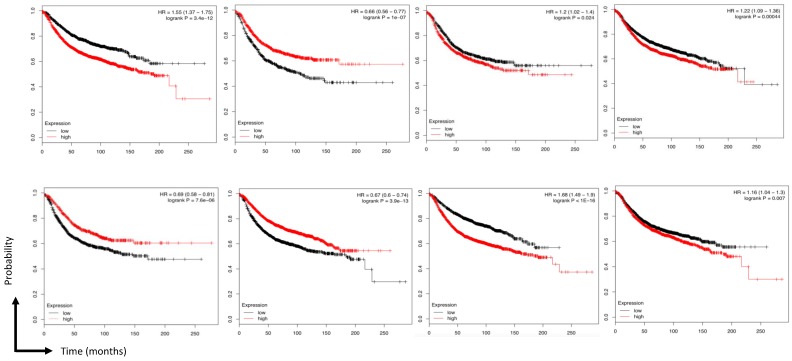
Survival analysis of eight hub genes in breast cancers. Overall survival analysis *TXN, ANXA2, TPM4, LOXL2, TPRN, ADCY6, TUBA1C and CMIP*. (Red lines represent the samples with a highly expressed gene and the blue line represents samples with a lowly expressed gene. HR: hazard ratio).

**Table 1 jcm-08-01160-t001:** The KEGG pathways in the enrichment analysis of *TXN*, *ANXA2*, *TPM4*, *LOXL2*, *TPRN*, *ADCY6*, *TUBA1C* and *CMIP*.

Gene Name	Subtype	KEGG Pathways
*TXN*	Nonmalignant	Fluid shear stress and atherosclerosis NOD-like receptor signaling pathway
*ANXA2*	Nonmalignant	No Data Available
*LOXL2*	Luminal	No Data Available
*TPM4*	Luminal	Adrenergic signaling in cardiomyocytes Cardiac muscle contraction Dilated cardiomyopathy (DCM) Hypertrophic cardiomyopathy (HCM)
*TUBA1C*	Basal	Apoptosis Gap junction Phagosome Tight junction
*CMIP*	Basal	No Data Available
*TPRN*	Claudin-low	No Data Available
*ADCY6*	Claudin-low	Adrenergic signaling in cardiomyocytes Dilated cardiomyopathy (DCM) Gap junction Aldosterone synthesis and secretion Apelin signaling pathway Bile secretion cAMP signaling pathway

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
