# Peer review of "Identification of Prognostic Candidate Genes in Breast Cancer by Integrated Bioinformatic Analysis"

_jcm, 2019, doi:10.3390/jcm8081160_

Round 1

Reviewer 1 Report

The issue is interesting; bioinformatic analysis could be an important tool in cancer research. However, the results are presented in an obscure way and the paper is difficult to understand  by a medical readership.

In particular:

Introduction section is prolix and often confuse; some phrases are unclear (for example, lined 60-63 and lines 71-72). The aim of the study is not adequately clarified.

Results section is also often unclear.

Discussion section contains general statements that are not relevant for the specific issue (for exapmle, lined 242-248). This section should underline and discuss the principal data emerging from the study. Authors should also hypothesize the possible applications of their results in disease management. Limitations of the study must be described at the end of this section.

Conclusions section is not pertinent. This section should briefly summarize the principal results and mention future directions.

Author Response

Thanks

Reviewer 2 Report

Charles C.N. Wang et.al. were intented to identifiy key genes as potential new biomarkers, using three different databases.  I did not have any major concerns on this article. I have a minor concern:  it would be more informative if the authors could be more specific concerning the molecular pathways that the eight key genes: TXN, ANXA2, TPM4, LOXL2, 293 TPRN, ADCY6, TUBA1C and CMIP are involved. Do these genes participate to the same pathways? Any interactions?

Author Response

Thanks

Round 2

Reviewer 1 Report

Authors have clarified most obscure points. I recommend an accurate English revision only

Author Response

We would like to thank the reviewer for accepting our reply, and we also revised the English according to the reviewer's suggestion. We believe that our manuscript is now more suitable for considering its publications. If you have any other questions, please let us know.